# Intrinsic Explanation of Random Subspace Method for Enhanced Security Applications

## ABSTRACT

Random subspace method has wide security applications such as providing certified defenses against adversarial and backdoor attacks, and building robustly aligned LLM against jailbreaking attacks. However, the explanation of random subspace methods lacks sufficient exploration. Existing state-of-the-art feature attribution methods such as Shapley value and LIME are computationally impractical and lack security guarantees when applied to random subspace methods. In this work, we propose EnsembleSHAP, an intrinsically faithful and secure feature attribution for random subspace methods that reuses its computational byproducts. Specifically, our feature attribution method is 1) computationally efficient, 2) maintains essential properties of effective feature attribution (such as local accuracy), and 3) offers guaranteed protection against attacks on feature attribution methods. We perform comprehensive evaluations for our explanation's effectiveness when faced with different empirical attacks. Our experimental results demonstrate that our explanation not only faithfully reports the most important features, but also certifiably detects the harmful features embedded in the input sample.

WARNING: This document may include content that could be considered offensive or harmful.

## 1 INTRODUCTION

Random subspace method (Ho, 1998) is widely employed for security purposes (Jia et al., 2021; Levine & Feizi, 2020; Robey et al., 2023; Zhang et al., 2023; Wang et al., 2021; Zeng et al., 2023; Cao et al., 2023), such as providing certified defenses (Levine & Feizi, 2020; Zeng et al., 2023; Zhang et al., 2023; Wang et al., 2021) against adversarial attacks, and enhancing the robustness of large language models against jailbreaking attacks (Cao et al., 2023; Robey et al., 2023). This method begins by generating predictions for multiple sub-sampled versions of a given input sample using a base model. It then creates an ensemble model that aggregates these predictions using a majority vote to determine the final prediction. As this approach only requires black-box access to the base model, it can be applied across different base model architectures (Levine & Feizi, 2020; Zeng et al., 2023; Zhang et al., 2023; Wang et al., 2021; Robey et al., 2023). Understanding the output of the random subspace method is crucial. For instance, in defending against jailbreaking attacks, it's essential for users to pinpoint the specific elements of the input prompt that lead to its classification as 'harmful' (or 'non-harmful'). Additionally, when a certified defense is compromised by strong empirical attacks, it becomes important for users to determine the specific adversarial words that caused the misclassification.

However, existing state-of-the-art feature attribution methods for black-box models (Lundberg & Lee, 2017; Chen et al., 2023b; Ribeiro et al., 2016; Enouen et al., 2023; Paes et al., 2024; Amara et al., 2024; Mosca et al., 2022; Lopardo et al., 2023) such as Shapley values (Lundberg & Lee, 2017; Chen et al., 2023b) and LIME (Ribeiro et al., 2016) have following disadvantages when applied to random subspace method. Firstly, they incur prohibitively high computational costs. Specifically, these methods involve randomly perturbing the input sample numerous times, denoted by $M$. The explained model then generates a prediction for each perturbed version of the input. In the case of the random subspace method, the objective is to explain the behavior of the ensemble model. Therefore, for each perturbed input version, $N$ sub-sampled variants of it are created and each is evaluated to generate the ensemble model's prediction, resulting in $M \times N$ total queries to the base model for just one input sample. In practical applications, both $N$ and $M$ can exceed $1,000$ (Enouen et al.,

2023; Zeng et al., 2023; Levine & Feizi, 2020), which significantly increases the computational cost. Secondly, current feature attribution methods do not provide security guarantees against the recently proposed *explanation-preserving attack* (Nadeem et al., 2023; Noppel & Wressnegger, 2023). In this type of attack, an adversary can perturb certain features of the input sample to cause misclassifications, and at the same time conceal these changes by preserving the original explanation.

**Our contribution.** In this work, we propose a computationally efficient feature attribution method for random subspace methods that is inherently faithful and secure. The key intuition is from the fact that the output of the ensemble model aggregates prediction outcomes of all sub-sampled inputs, and the impact of any specific sub-sampled input on the ensemble model's output can be further distributed to the individual features within that sub-sampled input. Consequently, we can infer the contribution of each feature from the predictions of all sub-sampled inputs that contain it, which are already calculated for producing the ensemble model's prediction.

We conduct a theoretical analysis to show that our method maintains key properties of Shapley value and is provably robust against explanation-perserving attacks on feature attribution methods. Additionally, we carry out empirical evaluations to assess the effectiveness of our explanations across various security applications of the feature attribution method.

## 2 BACKGROUND AND RELATED WORK

We begin by introducing the random subspace method and its security applications, followed by a discussion on existing feature attribution methods and their limitations.

### 2.1 RANDOM SUBSPACE METHOD

Random subspace-based method (Ho, 1998) has broad applications in security domains, such as certified defense mechanisms (Levine & Feizi, 2020; Zeng et al., 2023; Zhang et al., 2023; Wang et al., 2021) and protections against jailbreaking attacks (Robey et al., 2023; Cao et al., 2023). In particular, this method is agnostic to model architecture and scalable to large neural networks. Next, we summarize a general framework for the random subspace method and introduce its security applications.

**Building an ensemble model.** Suppose we have a testing input $\boldsymbol{x} = \{x_1, x_2, \cdots, x_d\}$ that consists of $d$ elements, where each element represents a feature of the input. For instance, when $\boldsymbol{x}$ is a text, each $x_i$ represents a word. We use $f : X \to [C]$ to represent the base model, where $X$ is the space of model input and $[C]$ represents the set of unique labels $\{1, 2, ..., C\}$ that base model can output. For example, in a binary classification problem, $[C] = \{1, 2\}$. For the simplicity of notation, we define a simplified base model $h$ that can take subsets of $\boldsymbol{x}$ as input. Specifically, we define $h : \mathcal{P}(\boldsymbol{x}) \to [C]$ as $h(\boldsymbol{z}) = f(\text{ABLATE}(\boldsymbol{x}, \boldsymbol{z}))$, where $\mathcal{P}(\boldsymbol{x})$ is the power set of $\boldsymbol{x}$, $\boldsymbol{z} \in \mathcal{P}(\boldsymbol{x})$ is a subset of $\boldsymbol{x}$ and ABLATE replaces all features of $\boldsymbol{x}$ not in $\boldsymbol{z}$ by a special value (e.g., the '[MASK]' token). That is, $x_i$

$$= \begin{cases} x_i & \text{if } x_i \in \boldsymbol{z} \\ SpecialValue & \text{otherwise} \end{cases}.$$

The random subspace method first uses the simplified base model $h$ to make predictions for random subsets of $\boldsymbol{x}$. Formally, we define the probability that a label $c \in [C]$ is predicted by the base model as:

$$p_c(\boldsymbol{x}, h, k) = \mathbb{E}_{\boldsymbol{z} \sim \mathcal{U}(\boldsymbol{x}, k)}[\mathbb{I}(h(\boldsymbol{z}) = c)], \tag{1}$$

where $\mathbb{I}$ is an indicator function whose output is 1 if the condition is satisfied and 0 otherwise, and $\boldsymbol{z} \sim \mathcal{U}(\boldsymbol{x}, k)$ is a subset of $\boldsymbol{x}$ with size $k$ that is randomly sampled from the uniform distribution. i.e., $\text{Pr}(\boldsymbol{z} = \boldsymbol{z}' \mid \boldsymbol{z} \sim \mathcal{U}(\boldsymbol{x}, k)) = \frac{1}{\binom{d}{k}}$ for any $\boldsymbol{z}' \subseteq \boldsymbol{x}$ satisfying $|\boldsymbol{z}'| = k$. Then the label with the largest probability is viewed as the predicted label of the ensemble classifier $H$ for the testing input $\boldsymbol{x}$, i.e.,

$$H(\boldsymbol{x}, h, k) = \arg\max_c p_c(\boldsymbol{x}, h, k). \tag{2}$$

In practice, the Random Subspace Method approximates the probability $p_c$ through Monte Carlo sampling. Initially, it generates $N$ groups of features from the original set $\boldsymbol{x}$ by sampling without replacement according to a uniform distribution $\mathcal{U}(\boldsymbol{x}, k)$. These subsets are represented as a collection of feature groups $G = \{\boldsymbol{z}_1, \ldots, \boldsymbol{z}_N\}$. For each of these feature groups $\boldsymbol{z}_j$, the method employs a

base classifier to predict a label. It then counts the occurrences $n_c$ of each possible label $c$ within a predetermined set of labels $1, 2, \ldots, C$, where $C$ represents the total number of unique labels. The calculation of $n_c$ is formally described by the equation:

$$n_c(\boldsymbol{x}, h, k) = \sum_{j=1}^{N} \mathbb{I}(h(\boldsymbol{z}_j) = c), c = 1, 2, \cdots, C, \tag{3}$$

where $\mathbb{I}$ denotes the indicator function, returning 1 when its condition is met and 0 otherwise. Consequently, the label probability $p_c(\boldsymbol{x}, h, k)$ is estimated by $\frac{n_c(\boldsymbol{x}, h, k)}{N}$.

**Security applications of existing random subspace method.** Random subspace method is used to build state-of-the-art certified defenses (Levine & Feizi, 2020; Zeng et al., 2023; Wang et al., 2021; Zhang et al., 2023). Many previous studies (Levine & Feizi, 2020; Zhang et al., 2023) showed that the ensemble model built by a random subspace method is certifiably robust against adversarial attacks, i.e., its prediction for a testing input remains unchanged once the $\ell_0$-norm perturbation to the testing input is bounded.

Another strand of research (Robey et al., 2023; Cao et al., 2023) uses the random subspace method to build robust LLM against jailbreaking attacks, leveraging the fragility of adversarially-generated jailbreaking prompts to perturbations. These methods first use the LLM to generate responses for each of the perturbed input prompts, and these responses are then labeled as either 'harmful' or 'non-harmful' by checking keywords. Lastly, these labels are aggregated to determine whether the input prompt should be approved or rejected.

Existing studies mainly focus on robustness, leaving the explanation of the random subspace method unexplored. Next, we introduce feature attribution to explain model outputs.

## 2.2 Feature Attribution

Feature attribution aims to explain why a machine learning model makes a certain prediction for an input by attributing the prediction to the most important features in the input. Existing feature attribution techniques (Lundberg & Lee, 2017; Chen et al., 2023b; Ribeiro et al., 2016; Paes et al., 2024; Mosca et al., 2022; Petsiuk et al., 2018; Sundararajan et al., 2017; Shrikumar et al., 2017; Smilkov et al., 2017) fall into two main categories: 1) white-box methods, exemplified by integrated gradients (Sundararajan et al., 2017) and DeepLIFT (Shrikumar et al., 2017), and 2) black-box methods, including LIME (Ribeiro et al., 2016) and Shapley values (Lundberg & Lee, 2017; Mosca et al., 2022; Chen et al., 2023b; Sundararajan & Najmi, 2020). White-box methods require knowledge of the explained model's architecture, parameters, and gradients, whereas black-box methods do not rely on such detailed knowledge. This study concentrates on black-box feature attribution methods due to their general applicability across various model architectures.

**Attacks to feature attribute methods.** Recent studies (Noppel & Wressnegger, 2023; Nadeem et al., 2023) have proposed the *explanation-preserving attack* to feature attribution methods. This attack involves adversarially perturbing the input sample in a manner that induces misclassifications while retaining the original explanation. This attack could be employed to conceal ongoing input manipulation (Zhang et al., 2020). For instance, an attacker could replace certain words in a clean sentence with adversarial alternatives, leading to misclassification, while those words still maintain low relevance in the resulting explanation.

**Limitations of existing feature attribute methods.** Existing state-of-the-art black-box feature attribution methods (Lundberg & Lee, 2017; Chen et al., 2023b; Ribeiro et al., 2016; Enouen et al., 2023; Paes et al., 2024; Amara et al., 2024; Mosca et al., 2022; Lopardo et al., 2023; Petsiuk et al., 2018) have following limitations. Firstly, they are computationally inefficient when applied to the random subspace method. Techniques such as LIME (Ribeiro et al., 2016) and Shapley values (Lundberg & Lee, 2017; Enouen et al., 2023; Chen et al., 2023b) necessitate a large number of queries (e.g., 1,000) to the black-box model using perturbed versions of the test input. As detailed in Section 2.1, each query to the ensemble classifier requires aggregating the prediction outcomes from all sub-sampled versions of the perturbed test input, leading to prohibitively high computation costs. Secondly, these methods lack theoretical guarantees regarding their performance when subjected to explanation-preserving attacks. In the next section, we design a feature attribution method that overcomes these limitations.

## 3 PROBLEM FORMULATION

### 3.1 FEATURE ATTRIBUTION FOR RANDOM SUBSPACE METHOD

Consider an ensemble model $H$ with base model $h$ and sub-sampling size $k$. For a given test input $\boldsymbol{x}$, consisting of $d$ features (denoted by $\boldsymbol{x} = \{x_1, x_2, \cdots, x_d\}$), let $\hat{y}$ represent the predicted label for $\boldsymbol{x}$, such that $H(\boldsymbol{x}, h, k) = \hat{y}$. The objective of feature attribution (Sundararajan et al., 2017; Paes et al., 2024; Amara et al., 2024; Lopardo et al., 2023; Chuang et al., 2024) is to assign an importance score $\alpha_i^{\hat{y}}$ to each element $x_i \in \boldsymbol{x}$, indicating its contribution to the ensemble model's prediction of $\hat{y}$. For instance, if $\boldsymbol{x}$ is a text consisting of $d$ words, each word would receive an importance score. By ranking these scores, users can easily identify the most influential words leading to the ensemble model's prediction.

### 3.2 DESIGN GOAL

Our approach is guided by three primary design goals. First, the feature attribution method should be computationally efficient, as predictions from an ensemble model are already resource-intensive, so the method must avoid repeatedly using the ensemble model for predictions. Second, it should adhere to key properties of effective feature attribution (Lundberg & Lee, 2017), such as local accuracy. Third, the method must be certifiably robust against explanation-preserving attacks. Specifically, if an adversary modifies a small number of features in the input to change the model's prediction, the most important features reported by the attribution method should include these adversarially altered features.

## 4 ALGORITHM DESIGN

Next, we introduce our EnsembleSHAP. Following existing feature attribution works (Sundararajan et al., 2017; Lopardo et al., 2023; Chuang et al., 2024), we explain for the model's prediction by measuring the contribution of each feature to the model's output label (denoted as $\hat{y}$). Specifically, we define the important score of the $i$-th feature for the predicted label $\hat{y}$ as:

$$\alpha_i^{\hat{y}}(\boldsymbol{x}, h, k) = \frac{1}{k} \mathbb{E}_{\boldsymbol{z} \sim \mathcal{U}(\boldsymbol{x}, k)}[\mathbb{I}(x_i \in \boldsymbol{z}) \cdot \mathbb{I}(h(\boldsymbol{z}) = \hat{y})]. \tag{4}$$

This importance score of a feature $x_i$ can be seen as the probability that a randomly sampled feature group contains $x_i$ and predicts for $\hat{y}$. The intuition behind this importance value is that the output generated by the ensemble model reflects the aggregated impact of all feature groups. For any given feature group $\boldsymbol{z}_j \in G$ having size $k$, the contribution of each feature to this group's result is equally divided, amounting to $\frac{1}{k}$ of the group's outcome. If a feature is not in a given group, then the contribution of this feature to this group's result is 0. Consequently, the contribution of a single feature is the aggregate of its contributions across all groups. This intuition leads to the property of local accuracy, which will be discussed in Section 5.

In practice, we use Monte Carlo sampling to approximate the importance score. We first sub-sample $N$ times to get a set of feature groups, denoted by $G = \{\boldsymbol{z}_1, \ldots, \boldsymbol{z}_N\}$, and get the base model's prediction for each of these feature groups. Then the importance score can be naively approximated as $\frac{1}{k \cdot N} \sum_{j=1}^{N} [\mathbb{I}(x_i \in \boldsymbol{z}_j) \cdot \mathbb{I}(h(\boldsymbol{z}_j) = \hat{y})]$. When the number of sub-sampled groups, denoted as $N$, is large, each feature is likely to appear in a similar number of groups. However, with a smaller $N$, variations in the appearance frequency can result in an unfair assessment of their importance. These features that appear more frequently in sub-sampled feature groups are likely to have greater importance. To solve this problem, we observe that the important score of feature $i$ for the predicted label $\hat{y}$ can be rewritten as:

$$\alpha_i^{\hat{y}}(\boldsymbol{x}, h, k) \tag{5}$$

$$= \frac{1}{k} \mathbb{E}_{\boldsymbol{z} \sim \mathcal{U}(\boldsymbol{x}, k)}[\mathbb{I}(x_i \in \boldsymbol{z}) \cdot \mathbb{I}(h(\boldsymbol{z}) = \hat{y})] \tag{6}$$

$$= \frac{1}{k} \Pr(x_i \in \boldsymbol{z}) \cdot \Pr(h(\boldsymbol{z}) = \hat{y} | x_i \in \boldsymbol{z}) \tag{7}$$

$$= \frac{1}{d} \Pr(h(\boldsymbol{z}) = \hat{y} | x_i \in \boldsymbol{z}). \tag{8}$$

Then the importance score can be approximated by:

$$\alpha_i^{\hat{y}}(\boldsymbol{x}, h, k) \approx \frac{1}{d \cdot \sum_{j=1}^{N} \mathbb{I}(x_i \in \boldsymbol{z}_j)} \sum_{j=1}^{N} \mathbb{I}(x_i \in \boldsymbol{z}_j) \cdot \mathbb{I}(h(\boldsymbol{z}_j) = \hat{y}), \tag{9}$$

where $d$ is the total number of features. The introduction of the new normalization term, $\sum_{j=1}^{N} \mathbb{I}(x_i \in \boldsymbol{z}_j)$, helps to mitigate the issue of unbalanced frequency.

**Computation cost.** Our method utilizes the predictions from the base model for each feature group $z_j \in G$, which are already computed for producing the prediction of the ensemble model. Therefore, our method adds negligible additional computational time.

## 5 THEORETICAL ANALYSIS

In this section, we begin by establishing the predicted label probability $p_{\hat{y}}$ on perturbed testing inputs to support subsequent theoretical analysis. Subsequently, we demonstrate that our method adheres to fundamental properties for effective feature attribution. Finally, we provide theoretical guarantees regarding our method's performance under attacks to feature attribution. For simplicity, we abuse the notation and use $i$ to represent the feature $x_i$ for theoretical analysis.

### 5.1 DEFINE $p_{\hat{y}}$ ON FEATURE SUBSETS

Before theoretical analysis, we first define the predicted label probability $p_{\hat{y}}$ when a subset of features $S \subseteq \boldsymbol{x}$ is present. In this case, random subspace method sub-samples feature groups with size $k$ from $S$. Particularly, given any feature subset $S$, we define the probability that the label $\hat{y}$ is predicted by the base model (when features not in $S$ are removed) as:

$$p_{\hat{y}}(S, h, k) = \mathbb{E}_{\boldsymbol{z} \sim \mathcal{U}(S,k)}[\mathbb{I}(h(\boldsymbol{z}) = \hat{y})], \tag{10}$$

where $\boldsymbol{z} \sim \mathcal{U}(S, k)$ is a subset of $S$ with size $k$ that is randomly sampled from the uniform distribution. i.e., $\Pr(\boldsymbol{z} = \boldsymbol{z}' \mid \boldsymbol{z} \sim \mathcal{U}(S, k)) = \frac{1}{\binom{|S|}{k}}$ for any $\boldsymbol{z}' \subseteq S$. We note that there is a special case when $|S| < k$. In this case, we let $p_{\hat{y}}(S, h, k) = \frac{1}{C}$, which means that the base model randomly guesses the label. We note that this assumption is necessary if we want to define Shapley value for random subspace method, because it is impossible to sub-sample $k$ features from less than $k$ features. In the following section, we utilize this definition to establish a Shapley value for random subspace method.

### 5.2 SHAPLEY VALUE BASED EXPLANATION FOR RANDOM SUBSPACE METHOD

Derived from game theory (Shapley et al., 1953), Shapley values are intended for credit assignment among players in cooperative games. A game is represented by a set of players $D$ and a value function $v(S) : \mathcal{P}(D) \to \mathbb{R}$, where $\mathcal{P}(D)$ means the power set of $D$. The Shapley value for player $i$ is defined as:

$$\phi_i(v) = \sum_{S \subseteq D \setminus \{i\}} \frac{|S|!(d - |S| - 1)!}{d!} (v(S \cup \{i\}) - v(S)). \tag{11}$$

Shapley value has long been regarded as the gold standard for feature attribution (Lundberg & Lee, 2017; Mosca et al., 2022; Chen et al., 2023b; Sundararajan & Najmi, 2020; Paes et al., 2024; Amara et al., 2024; Sundararajan et al., 2017). In order to explain the output of a machine learning model, many existing works (Paes et al., 2024; Amara et al., 2024; Sundararajan et al., 2017) use the probability of the the model's output as the value function. Similarly, we can define a Shapley value for random subspace method. Specifically, we let the label probability $p_{\hat{y}}$ be the value function $v$ and let the input feature set $\boldsymbol{x}$ be the set of players $D$. Then the Shapley value for feature $i$ can be written as:

$$\phi_i(p_{\hat{y}}) = \sum_{S \subseteq \boldsymbol{x} \setminus \{i\}} \frac{|S|!(d - |S| - 1)!}{d!} (p_{\hat{y}}(S \cup \{i\}, h, k) - p_{\hat{y}}(S, h, k)). \tag{12}$$

This value is empirically challenging to compute because $p_{\hat{y}}$ should be evaluated on all feature subsets, while evaluating $p_{\hat{y}}$ on a single feature subset requires $N$ forward passes of the base model. In the next part, we demonstrate that our computationally efficient importance score maintains the key properties of Shapley value.

## 5.3 PROPERTIES OF ENSEMBLESHAP

EnsembleSHAP possesses *local accuracy* and *symmetry* as derived from Shapley values (**?**Chen et al., 2023a), whilst substituting the remaining two properties inherent to Shapley values, specifically *dummy* and *linearity*, with *order consistency* (with Shapley value). The linearity property is omitted because its application is not straightforward in the context of subspace methods. Furthermore, the relaxation of the dummy property is from the observation that in many cases, people are more interested in the comparative importance of features over their absolute importance scores (Lopardo et al., 2023; Xue et al., 2024). We introduce these properties below.

The first property is *local accuracy*. This property ensures that the explanation accurately reflects the behavior of the ensemble model for the testing input $\boldsymbol{x}$. It can be formally stated as follows.

**Property 1.** *(Local accuracy). For any $\boldsymbol{x}$, $h$, and $k$, the importance score of all features sum up to $p_{\hat{y}}(\boldsymbol{x}, h, k)$, i.e., $\sum_{i \in \boldsymbol{x}} \alpha_i^{\hat{y}}(\boldsymbol{x}, h, k) = p_{\hat{y}}(\boldsymbol{x}, h, k)$.*

The second property is *symmetry*. The symmetry property states that if two features contribute equally to all possible feature subsets $S \subseteq \boldsymbol{x}$, then feature $i$ and $j$ should receive the same importance score.

**Property 2.** *(Symmetry). Given a pair of features $(i, j)$, if for any $S \subseteq \boldsymbol{x} \setminus \{i, j\}$, $p_{\hat{y}}(S \cup \{i\}, h, k) = p_{\hat{y}}(S \cup \{j\}, h, k)$, then $\alpha_i^{\hat{y}}(\boldsymbol{x}, h, k) = \alpha_j^{\hat{y}}(\boldsymbol{x}, h, k)$.*

The third property is *order consistency* (with Shapley value). This property ensures that if Shapley value ranks a feature as more significant, our attribution approach will also give it a higher importance. The Shapley value for random subspace method is defined in Section 5.2.

**Property 3.** *(Order consistency with Shapley value). Given a pair of features $(i, j)$, $\alpha_i^{\hat{y}}(\boldsymbol{x}, h, k) \geq \alpha_j^{\hat{y}}(\boldsymbol{x}, h, k)$ if and only if $\phi_i(p_{\hat{y}}) \geq \phi_j(p_{\hat{y}})$, where $\phi_i(p_{\hat{y}})$ and $\phi_j(p_{\hat{y}})$ respectively represent Shapley values of $i$ and $j$.*

We provide the proof details in Appendix A. Our method essentially relaxes the dummy property of Shapley value to simplify its complex computation. Despite this alteration, the utility of the Shapley value is preserved in most scenarios due to the property of order consistency. This observation is supported by these commonly used metrics for feature attribution, such as the fidelity score (Miró-Nicolau et al., 2024; Chuang et al., 2024), perturbation curves (Paes et al., 2024; Chen et al., 2020) and faithfulness (Lopardo et al., 2023). These metrics rely on the relative order of importance scores rather than their absolute values.

## 5.4 CERTIFIED DETECTION OF ADVERSARIAL FEATURES

In this part, we demonstrate that our explanation method provably detects adversarial features that causes model misclassifiation, therefore is provably secure against *explanation-preserving attacks*. We suppose the attacker can modify at most $T$ features of the original testing input $\boldsymbol{x}$ to change the predicted label of the ensemble classifier. We denote the set of all possible perturbed test inputs $\boldsymbol{x}'$ as $\mathcal{B}(\boldsymbol{x}, T)$, and we use $\boldsymbol{x} \ominus \boldsymbol{x}'$ to denote the set of modified features. Here, we focus on top-$e$ most important features reported by our method. i.e., $e$ features with highest importance scores for the predicted label. We denote this set of features before attack as $E(\boldsymbol{x})$, and use $E(\boldsymbol{x}')$ to represent the new set of top-$e$ most important features for $\boldsymbol{x}'$. Our goal is to derive the *certified detection size* $\mathcal{D}(\boldsymbol{x}, T)$, which is the intersection size lower bound between the set of modified features and the set of reported important features, which is formally defined as:

$$\mathcal{D}(\boldsymbol{x}, T) = \arg\max_{r}, s.t. |(\boldsymbol{x}' \ominus \boldsymbol{x}) \cap E(\boldsymbol{x}')| \geq r, \forall \boldsymbol{x}' \in \mathcal{B}(\boldsymbol{x}, T), H(\boldsymbol{x}') \neq H(\boldsymbol{x}). \quad (13)$$

We have the following result:

**Theorem 1.** *Given a testing input $\boldsymbol{x}$ which is originally predicted as $\hat{y}$. We suppose there exists $\boldsymbol{x}' \in \mathcal{B}(\boldsymbol{x}, T)$ such that $H(\boldsymbol{x}') \neq \hat{y}$. Then $\mathcal{D}(\boldsymbol{x}, T)$ is the solution of the following optimization*

*problem:*

$$\mathcal{D}(\boldsymbol{x}, T) = \arg\max_r r, \ s.t. \ \forall \hat{y}' \neq \hat{y}, \tag{14}$$

$$\overline{\alpha}^{\hat{y}'}_{w_{e-r+1}}(\boldsymbol{x}, h, k) + \frac{1}{d} - \frac{1}{k}\frac{\binom{d-1-T}{k-1}}{\binom{d}{k}} \tag{15}$$

$$\leq \frac{1}{T-r+1}\Big[\frac{1}{2k} \cdot (\underline{p}_{\hat{y}}(\boldsymbol{x}, h, k) - \overline{p}_{\hat{y}'}(\boldsymbol{x}, h, k)) - \frac{r-1}{d} + \sum_{i=d-T+r}^{d} \underline{\alpha}^{\hat{y}'}_{q_i}(\boldsymbol{x}, h, k)\Big] \tag{16}$$

$$\vee \tag{17}$$

$$\frac{1}{e-r+1}\sum_{i=1}^{e-r+1} \overline{\alpha}^{\hat{y}'}_{w_i}(\boldsymbol{x}, h, k) - \frac{1}{T-r+1}\sum_{i=d-T+r}^{d} \underline{\alpha}^{\hat{y}'}_{q_i}(\boldsymbol{x}, h, k) + \frac{r-1}{d \cdot (T-r+1)} \tag{18}$$

$$\leq \frac{1}{2k}\Big(\frac{1}{T-r+1} - \frac{k-1}{e-r+1}\Big) \cdot (\underline{p}_{\hat{y}}(\boldsymbol{x}, h, k) - \overline{p}_{\hat{y}'}(\boldsymbol{x}, h, k)), \tag{19}$$

*where $\underline{p}_c$ (or $\overline{p}_c$) represents the probability lower (or upper) bound of some label $c \in [C]$, $\underline{\alpha}^{\hat{y}'}_i(\boldsymbol{x}, h, k)$ (or $\overline{\alpha}^{\hat{y}'}_i(\boldsymbol{x}, h, k)$) represents the lower (or upper) bound of the feature $i$'s importance score for some label $\hat{y}' \neq \hat{y}$, $\{w_1, \cdots, w_d\}$ denotes the set of all features in descending order of the important value upper bound $\overline{\alpha}^{\hat{y}'}(\boldsymbol{x}, h, k)$, i.e., $\overline{\alpha}^{\hat{y}'}_{w_1}(\boldsymbol{x}, h, k) \geq \overline{\alpha}^{\hat{y}'}_{w_2}(\boldsymbol{x}, h, k) \geq \cdots \geq \overline{\alpha}^{\hat{y}'}_{w_d}(\boldsymbol{x}, h, k)$, and $\{q_1, \cdots, q_d\}$ denotes the set of all features in descending order of the important value lower bound $\underline{\alpha}^{\hat{y}'}(\boldsymbol{x}, h, k)$, i.e., $\underline{\alpha}^{\hat{y}'}_{q_1}(\boldsymbol{x}, h, k) \geq \underline{\alpha}^{\hat{y}'}_{q_2}(\boldsymbol{x}, h, k) \geq \cdots \geq \underline{\alpha}^{\hat{y}'}_{q_d}(\boldsymbol{x}, h, k)$.*

The specifics for computing $\underline{p}_{\hat{y}}$, $\overline{p}_{\hat{y}'}$, $\underline{\alpha}^{\hat{y}'}_i(\boldsymbol{x}, h, k)$, and $\overline{\alpha}^{\hat{y}'}_i(\boldsymbol{x}, h, k)$ can be found in Appendix C, and the proof is available in Appendix B. The underlying idea of this theorem is that to change the label from $\hat{y}$ to $\hat{y}'$, the attacker must ensure that more feature groups predict for $\hat{y}'$. However, the attacker can only alter the predicted labels of feature groups that include at least one feature in $\boldsymbol{x} \ominus \boldsymbol{x}'$. Consequently, the importance values of features within $\boldsymbol{x} \ominus \boldsymbol{x}'$ are likely to increase, making them more detectable.

# 6 EVALUATION ON SECURITY APPLICATIONS

We evaluate the effectiveness of our feature attribution method in the context of certified defense and defense against jailbreaking attacks, which are two critical security applications of random subspace method. For certified defense, we first test our approach's efficacy in scenarios without any attacks. Subsequently, we explore its performance under conditions where the certified defense can be compromised by powerful empirical attacks. Specifically, we employ a backdoor attack (BadNets (Gu et al., 2017)) and an adversarial attack (TextFooler (Jin et al., 2020)) to challenge the random subspace method (more details are provided in Appendix D.3). Our findings reveal that, even in instances where these attacks lead the ensemble model to incorrect predictions, our method successfully identifies the exact words responsible for the failure. For defense against jailbreaking attacks, we evaluate three types of such attacks: GCG (Zou et al., 2023), AutoDAN (Liu et al., 2023), and DAN (Liu et al., 2023). Our results demonstrate that our method is capable of identifying the harmful query embedded within the jailbreaking prompt.

## 6.1 EXPERIMENTAL SETUP

**Random Subspace Method Implementation.** For certified defense, we follow RanMASK (Zeng et al., 2023) for constructing the ensemble classifier. For defense against jailbreaking attacks, we adopt the RA-LLM (Cao et al., 2023) framework. More details are provided in Appendix D.4.

**Datasets.** We use classification datasets such as SST-2 (Socher et al., 2013), IMDB (Maas et al., 2011), and AGNews (Zhang et al., 2015) for the study on certified defense mechanisms, and use harmful behaviors dataset (Zou et al., 2023) for defense against jailbreaking attacks. More details can be found in Appendix D.1.

**Models.** For certified defense, we use a pretrained BERT model (Devlin et al., 2018) as our base model and fine-tune it using AdamW optimizer for 10 epochs on masked training samples to improve

the certification performance. The learning rate is set to $1 \times 10^{-5}$. For defense against jailbreaking attacks, we directly use Vicuna-7B (Chiang et al., 2023) as our base model.

**Hyper-parameters.** Unless specifically mentioned, we use following hyperparameters by default. For certified defense, the dropping rate (expressed as $\rho = 1 - \frac{k}{d}$) is set to $0.8$, and $N$ is set to $1,000$. For defense against jailbreaking attacks, we set the dropping rate to $0.4$, $N$ to $500$, and the threshold $\tau$ to $0.1$. The impact of these hyperparameters will be explored in an ablation study.

**Evaluation Metrics.** We use the following metrics to evaluate the effectiveness of our feature attribution method. The faithfulness metric is reported across all our experiments. Furthermore, in instances where there is ground-truth information regarding the key words that significantly influence the prediction of the ensemble model (e.g., during empirical attacks such as backdoor attacks), we implement extra metrics for predicting these key words. We denote the test dataset by $\mathcal{D}_{test}$, the base model by $h$ and the the prediction of the ensemble for some test sample $\boldsymbol{x}$ by $H(\boldsymbol{x})$.

- **Faithfulness (Lopardo et al., 2023).** We define the faithfulness of the feature attribution as the percentage of label flips when the $e$ features with highest importance scores are deleted. We use $E(\boldsymbol{x})$ to denote the $e$-most important features reported by the feature attribution method. Then faithfulness can be represented by:

$$Faithfulness = \frac{1}{|\mathcal{D}_{test}|} \sum_{\boldsymbol{x} \in \mathcal{D}_{test}} \mathbb{I}[H(\boldsymbol{x}) \neq H(\boldsymbol{x} \setminus E(\boldsymbol{x}))] \tag{20}$$

  In our experiment, we explore the impact of removing various percentages of important words by adjusting the ratio $\frac{e}{d}$ to different fractional values.

- **Key word prediction.** We define a set of ground-truth important words denoted by $L(\boldsymbol{x})$. In the context of a backdoor attack, $L(\boldsymbol{x})$ comprises the triggers that are inserted. For adversarial attacks, it includes the words that have been substituted. And in a jailbreaking attack, it consists of the harmful query embedded within the jailbreaking prompt. We let the feature attribution method identify the top $e$ most crucial words and measure the intersection of these words with the set of ground-truth important words. Specifically, we have *top-e precision*$= \frac{|E(\boldsymbol{x}) \cap L(\boldsymbol{x})|}{e}$, *top-e recall*$= \frac{|E(\boldsymbol{x}) \cap L(\boldsymbol{x})|}{|L(\boldsymbol{x})|}$, and *top-e f1-score*$= \frac{2 \cdot |E(\boldsymbol{x}) \cap L(\boldsymbol{x})|}{|L(\boldsymbol{x})| + e}$. As our final result, we report the average values of *top-e precision*, *top-e recall*, and *top-e f1-score* computed on $\mathcal{D}_{test}^*$ for different $e$ values. $\mathcal{D}_{test}^*$ is a specific subset of $\mathcal{D}_{test}$ detailed in Appendix D.5.

- **Certified detection rate.** We develop metrics for provable defense against explanation-preserving attacks discussed in Section 5.4. We define *certified detection rate* as $\mathcal{D}(\boldsymbol{x}, T)/T$ to measure the percentage of detected adversarial features. We report the mean values of certified detection rate computed on $\mathcal{D}_{test}$ for different $e$ values.

**Compared Methods.** We compare our method with following baseline methods. Shapley value (Chen et al., 2023b) and LIME (Ribeiro et al., 2016) are state-of-the-art techniques in feature attribution but present computational challenges when applied directly to ensemble models. Consequently, we implement these methods on the **base model**, anticipating that the ensemble model will exhibit similar behaviors. Furthermore, we have adapted the ICL method (Kroeger et al., 2023) for feature attribution purposes. This approach leverages the in-context learning capabilities of large language models (LLMs). We provide implementation details of these methods in Appendix D.2.

## 6.2 EXPERIMENTAL RESULTS

In this section, we present our evaluation results. We start by empirically comparing the explanation quality of our method against baselines for certified defense and defense against jailbreaking attacks. Following this, we assess the certified detection rate of adversarial features.

### 6.2.1 EXPLAIN CERTIFIED DEFENSE

We evaluate our method's explanation effectiveness both in the absence of attacks and in scenarios where the certified defense is compromised by strong empirical attacks.

**No Attack.** In Table 1, we present a comparison of our method's faithfulness against other baseline methods for clean test samples. We can see that our method surpasses all baselines in performance. For instance, within the IMDb dataset, our approach achieves a faithfulness rate of 60%, in contrast

**Table 1: Compare the faithfulness of our method with baselines for certified defense. We delete a certain percentage of most important words and compute the rate of label changes.**

| Defense scenarios | Dataset | SST-2 | | IMDb | | AG-news | |
|---|---|---|---|---|---|---|---|
| | Deletion ratio | 10% | 20% | 10% | 20% | 10% | 20% |
| No attack | Shapley value | 0.320 | 0.530 | 0.300 | 0.330 | 0.150 | 0.280 |
| | LIME | 0.125 | 0.145 | 0.060 | 0.095 | 0.020 | 0.035 |
| | ICL | 0.095 | 0.135 | 0.045 | 0.050 | 0.030 | 0.040 |
| | Ours | **0.365** | **0.605** | **0.600** | **0.745** | **0.175** | **0.410** |
| Backdoor attack | Shapley value | 0.380 | 0.630 | 0.520 | 0.540 | 0.725 | 0.790 |
| | LIME | 0.080 | 0.095 | 0.120 | 0.180 | 0.205 | 0.300 |
| | ICL | 0.055 | 0.085 | 0.120 | 0.170 | 0.140 | 0.235 |
| | Ours | **0.400** | **0.655** | **0.810** | **0.910** | **0.735** | **0.795** |
| Adversarial attack | Shapley value | 0.600 | 0.840 | 0.845 | 0.840 | 0.850 | 0.960 |
| | LIME | 0.100 | 0.160 | 0.280 | 0.335 | 0.200 | 0.265 |
| | ICL | 0.130 | 0.170 | 0.305 | 0.365 | 0.115 | 0.130 |
| | Ours | **0.680** | **0.880** | **0.980** | **1.000** | **0.905** | **0.970** |

**Table 2: Compare the key word prediction performance of our method with baselines for certified defense. Each feature attribution method reports the top-5 important words ($e = 5$).**

| Defense scenarios | Dataset | SST-2 | | | IMDb | | | AG-news | | |
|---|---|---|---|---|---|---|---|---|---|---|
| | Metric | Precision | Recall | F-1 score | Precision | Recall | F-1 score | Precision | Recall | F-1 score |
| Backdoor attack | Shapley value | 0.543 | 0.904 | 0.679 | 0.295 | 0.491 | 0.368 | 0.523 | 0.872 | 0.654 |
| | LIME | 0.148 | 0.247 | 0.185 | 0.037 | 0.022 | 0.027 | 0.073 | 0.122 | 0.091 |
| | ICL | 0.087 | 0.145 | 0.109 | 0.030 | 0.049 | 0.037 | 0.068 | 0.113 | 0.085 |
| | Ours | **0.585** | **0.975** | **0.731** | **0.535** | **0.892** | **0.669** | **0.557** | **0.929** | **0.697** |
| Adversarial attack | Shapley value | 0.361 | 0.680 | 0.433 | 0.282 | 0.142 | 0.159 | 0.528 | 0.343 | **0.352** |
| | LIME | 0.146 | 0.319 | 0.184 | 0.067 | 0.025 | 0.033 | 0.242 | 0.128 | 0.150 |
| | ICL | 0.098 | 0.210 | 0.122 | 0.076 | 0.040 | 0.042 | 0.080 | 0.046 | 0.050 |
| | Ours | **0.378** | **0.717** | **0.458** | **0.384** | **0.184** | **0.211** | **0.530** | **0.356** | 0.351 |

to the 30% attained by the Shapley value method when the deletion ratio is 10%. A visualization of our feature attribution for IMDb dataset is provided by Figure 7 in the Appendix.

**Backdoor Attack and Adversarial Attack.** Table 5 in Appendix shows that a significant proportion of testing samples can be compromised when the attacker could maliciously insert (or alter) a relatively large number of words. Table 1 details our method's capability in explaining model behavior to sentences altered by the backdoor (or adversarial) attack. Specifically, for the adversarial attack on the IMDb dataset, removing the 10% of words considered most critical by our method results in a label change for 98% of the adversarial sentences.

Additionally, Table 2 and Table 6 (in Appendix) provides evidence that, in scenarios where these altered sentences misguide the ensemble model towards incorrect predictions, our method exhibits superior capability in detecting the backdoor triggers (or adversarial words) responsible for the ensemble model's erroneous behavior. For example, on IMDB dataset, our technique achieves a recall of $0.892$, significantly higher than the $0.491$ recall obtained using Shapley value for backdoor attacks when $e = 5$. For a qualitative comparison, please see Figure 6 and Figure 7 in the Appendix.

### 6.2.2 EXPLAIN DEFENSE AGAINST JAILBREAKING ATTACKS

In this part, we demonstrate that our method enhances understanding of the decision-making processes of the RA-LLM (Cao et al., 2023) when faced with jailbreaking prompts. Table 3 shows that our method outperforms baselines in identifying the most important words that influence the RA-LLM's decisions for jailbreaking prompts. Table 4 and Table 7 (in Appendix) demonstrates that when a jailbreaking prompt is detected as 'harmful' by RA-LLM, our method is capable of identifying the harmful query embedded within the jailbreaking prompt that leads to this decision. This finding is also supported by the qualitative results shown in Figure 9 in Appendix.

### 6.2.3 IMPACT OF HYPERPARAMETERS

We examine how the dropping rate $\rho$ and the number of sub-sampled inputs $N$ influence our method's faithfulness and key word prediction performance. Figures 10 and 11 in Appendix demonstrates that both metrics generally improves with an increase in $N$, as it leads to a more precise estimation of importance values. Furthermore, Figures 12 and 13 in Appendix reveals that while key word

**Table 3: Evaluate the faithfulness of our method for defense against jailbreaking attacks. We delete a certain percentage of most important words and compute the rate of label changes.**

| Attack method | GCG | | AutoDAN | | DAN | |
|---|---|---|---|---|---|---|
| Deletion ratio | 10% | 20% | 10% | 20% | 10% | 20% |
| Shapley value | 0.11 | 0.19 | 0.15 | 0.18 | 0.33 | 0.33 |
| LIME | **0.15** | 0.23 | 0.34 | 0.32 | 0.54 | 0.38 |
| ICL | 0 | 0 | 0.08 | 0.11 | 0.24 | 0.27 |
| Ours | **0.15** | **0.24** | **0.38** | **0.46** | **0.85** | **0.74** |

**Table 4: Compare the key word prediction performance of our method with baselines for defense against jailbreaking attacks. Each feature attribution method reports the top-10 important words ($e = 10$).**

| Attack method | GCG | | | AutoDAN | | | DAN | | |
|---|---|---|---|---|---|---|---|---|---|
| Metric | Precision | Recall | F-1 score | Precision | Recall | F-1 score | Precision | Recall | F-1 score |
| Shapley value | 0.651 | 0.571 | 0.602 | 0.306 | 0.260 | 0.277 | 0.137 | 0.119 | 0.126 |
| LIME | 0.654 | 0.575 | 0.605 | 0.335 | 0.281 | 0.302 | 0.332 | **0.289** | **0.306** |
| ICL | 0.544 | 0.466 | 0.492 | 0.252 | 0.212 | 0.227 | 0.078 | 0.064 | 0.070 |
| Ours | **0.664** | **0.584** | **0.615** | **0.434** | **0.379** | **0.400** | **0.378** | 0.287 | **0.306** |

prediction performance remains stable, there is a decline in faithfulness at a very large $\rho$ value (e.g., $\rho = 0.9$). This is because the ensemble model becomes insensitive to the deletion of important features at higher dropping rates. We have consistent findings for defense against jailbreaking attacks, as illustrated in Figure 14 in Appendix.

## 6.3 Certified Detection of Adversarial Features

We evaluate the certified detection rate of our feature attribution on text classification datasets. By default, we set the certification confidence $1 - \beta$ to 0.99, the dropping rate $\rho$ to 0.8, and the sub-sampling number $N$ to 10,000. Figure 1 shows the results in default setting. The findings indicate that the certified detection rate improves as the explanation reports more features as important features, and the rate decreases when the attacker is able to modify a greater number of features. This decrease occurs because the attacker can make several perturbed features contribute to the target label simultaneously, making each individual feature less noticeable. Figure 15, Figure 16, and Figure 17 in Appendix shows the impact of $\beta$, $N$ and $\rho$, respectively. We find that while the certified detection rate is insensitive to the $\beta$ value, it can be significantly enhanced by increasing $N$, or $\rho$.

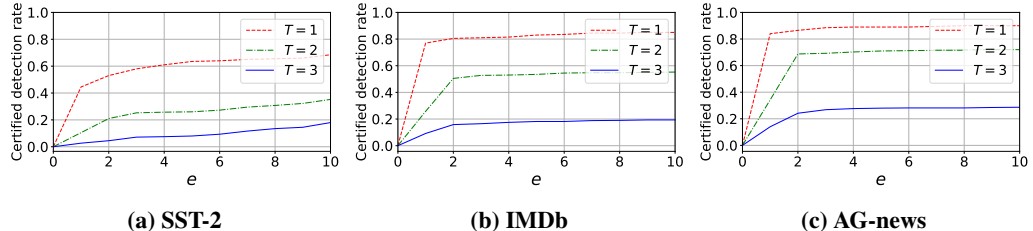

(a) SST-2      (b) IMDb      (c) AG-news

**Figure 1: Certified detection rate on text classification datasets. $T$ is the number of modified features, and $e$ is the number of reported most important features.**

## 7 Conclusion and Future Work

In this work, we propose an efficient feature attribution method for the random subspace method that is provably secure against explanation-preserving attacks. Potential future directions include: 1) investigating the explanation of random subspace method for privacy applications, such as machine unlearning (Bourtoule et al., 2021) and differential privacy (Liu et al., 2020); and 2) developing provably secure feature attribution methods for general machine learning models.

## 8 ETHICS STATEMENT

Our proposed method, EnsembleSHAP, provides a secure and efficient feature attribution for the random subspace method, and helps build ethical and explainable ML models. Our method can be applied to security-sensitive applications such as defending against adversarial and backdoor attacks and building robust language models (LLMs) resistant to jailbreaking attacks. By providing explanations for model decisions, we aim to enhance users' trust to AI systems. Nonetheless, this also means that practitioners must take responsibility for how these explanations are communicated to end users, ensuring that they are not misleading or overly simplified.

## 9 REPRODUCIBILITY STATEMENT

To ensure the reproducibility of our results, we have carefully designed our experiments using publicly available models and datasets. This allows other researchers to easily access the same resources and replicate our findings. In the evaluation section of the paper, we provide all hyperparameter settings for both certified defense scenarios and jailbreaking attack scenarios. Furthermore, we have included detailed hyperparameter settings for all attack methods in Appendix D, ensuring that the reproduction of adversarial attack experiments is fully transparent. Implementation details for each baseline explanation method are also provided in the same appendix, enabling researchers to precisely replicate the conditions under which the attacks were tested.

For the theoretical aspects of our work, we have included all proofs supporting our claims and properties in Appendix A (for the feature attribution properties) and Appendix B (for certified detection of adversarial features), providing a rigorous mathematical foundation for our contributions. This ensures that others can verify the correctness of the theory for our method. Finally, we commit to releasing our code upon paper acceptance.

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
