# OpenReview forum: "Intrinsic Explanation of Random Subspace Method for Enhanced Security Applications"
_ICLR.cc/2025/Conference — Submitted to ICLR 2025_

### Official Review · Reviewer_r65P · 2024-10-30

**Soundness:** 3
**Presentation:** 3
**Contribution:** 3
**Rating:** 6
**Confidence:** 3

**Summary:**

This work reveals two major issues with current state-of-the-art feature attribution methods: (1) high computational costs and (2) a lack of security guarantees against explanation-preserving attacks. To address these issues, this study proposes a computationally efficient and inherently secure feature attribution method. The key insight derives from the fact that an ensemble model’s output aggregates the prediction results of all sub-sampled inputs, with each sub-sampled input’s influence on the ensemble output further distributable to the individual features within that input. Thus, the contribution of each feature can be inferred from analyzing the prediction results of all sub-sampled inputs containing that feature.

**Strengths:**

A security guarantee is proposed for the explanation-preserving attack without increasing the high computational cost.

**Weaknesses:**

Adaptive attack discussion: Discussion and experiments on adaptive attacks could further strengthen the paper. If attackers know the defense strategy, what happens? For instance, they could adjust the attack target so that triggers do not fall within the top 10% or 20% of important features but rather within the top 30% or 40% to circumvent defenses.

**Questions:**

1. The two works compared by the authors are not defense-oriented, so is this comparison fair? Should comparisons also include existing defenses against backdoor and adversarial attacks for large language models (LLMs) to better evaluate the proposed method’s effectiveness?
2. Without prior knowledge, if the proposed method is used to defend and 10% or 20% of the important words are deleted, can the LLM still make accurate responses? The experimental results do not indicate whether the defense proposed in this paper affects the model's responses to normal text.
3. Regarding the faithfulness comparison in Table 1: Faithfulness is defined as the percentage of label flips when the top e features with the highest importance scores are deleted. My understanding is that this metric should be as high as possible under attack, as the deleted important features likely contain adversarial elements. In the absence of an attack, if deleting these features leads to a high label flip rate, it indicates that removing important features significantly impacts model performance. How should one decide whether or not to delete these features?
4. It is recommended that the authors add a discussion on adaptive attacks to enhance the practical value of the proposed method.

---

> ### Author Response · Authors · 2024-11-21
>
> We thank the reviewer for the constructive comments.
>
> **Weakness 1. Adaptive attack discussion: Discussion and experiments on adaptive attacks could further strengthen the paper. If attackers know the defense strategy, what happens? For instance, they could adjust the attack target so that triggers do not fall within the top 10% or 20% of important features but rather within the top 30% or 40% to circumvent defenses.**
>
> Thank you for the constructive suggestion. We think that designing adaptive attacks capable of simultaneously bypassing both the defense mechanism (the random subspace method) and staying undetected by our method is an intriguing and meaningful direction for future research.
>
> **Question 1.
> The two works compared by the authors are not defense-oriented, so is this comparison fair? Should comparisons also include existing defenses against backdoor and adversarial attacks for large language models (LLMs) to better evaluate the proposed method’s effectiveness?**
>
> We emphasize that our method is not primarily defense-oriented. Instead, we first establish that it provides a faithful explanation by proving its order consistency with the Shapley value. Faithfulness, in this context, implies that a feature with greater influence on the model's prediction is assigned a higher importance score. Consequently, the certified detection guarantee can be seen as a byproduct of this faithfulness. Moreover, as demonstrated in Table 1 of the experimental section, our method outperforms baseline approaches even when no attacks are present.
>
> While the random subspace method can be employed to defend against backdoor and adversarial attacks on LLMs, existing research has only focused on its use for jailbreaking attacks (which can be considered a more powerful form of adversarial attack without constraints on the perturbation boundary). We believe that exploring the application of the random subspace method for defending against backdoor and adversarial attacks on LLMs presents an intriguing direction for future research.
>
> **Question 2. Without prior knowledge, if the proposed method is used to defend and 10% or 20% of the important words are deleted, can the LLM still make accurate responses? The experimental results do not indicate whether the defense proposed in this paper affects the model's responses to normal text.**
>
> Our method serves as a post-hoc explanation tool for existing defense mechanisms that uses the random subspace method. It does not alter the decisions made by these defense mechanisms but instead identifies the most important features influencing those decisions.
>
> **Question 3. Regarding the faithfulness comparison in Table 1: Faithfulness is defined as the percentage of label flips when the top e features with the highest importance scores are deleted. My understanding is that this metric should be as high as possible under attack, as the deleted important features likely contain adversarial elements. In the absence of an attack, if deleting these features leads to a high label flip rate, it indicates that removing important features significantly impacts model performance. How should one decide whether or not to delete these features?**
>
> Thank you for the question. In this paper, faithfulness is employed as a metric to evaluate the performance of the explanation. It’s important to note that our method does not actually delete features in practice. Instead, it provides an explanation for the outcome of a defense mechanism (one using the random subspace method). For instance, when combined with RA-LLM in the context of jailbreaking attacks against LLMs, our method serves as a post-attack analysis tool. RA-LLM determines whether an attack has occurred, and our method is subsequently used to identify which parts of the input leads to that decision.
>
> **Question 4.
> It is recommended that the authors add a discussion on adaptive attacks to enhance the practical value of the proposed method.**
>
> Thank you for the valuable suggestion. We believe that developing adaptive attack strategies that can effectively bypass both the defense and the explanatory mechanisms presents an exciting and important avenue for future research.

---

### Official Review · Reviewer_52Co · 2024-11-02

**Soundness:** 1
**Presentation:** 3
**Contribution:** 2
**Rating:** 3
**Confidence:** 3

**Summary:**

This paper introduces EnsembleSHAP, a novel feature attribution method designed for random subspace methods. Compared with existing feature attribution techniques like Shapley values or LIME, the proposed EnsembleSHAP is both computationally efficient and intrinsically secure. Moreover, it provides a certified defense against various attacks.

**Strengths:**

1.	The proposed EnsembleSHAP addresses the security gap in existing feature attribution methods, offering certified defenses.
2.	The authors carry out empirical evaluations to assess the effectiveness of their explanations across various security applications of the feature attribution method.

**Weaknesses:**

1.	In section 4,  the importance scores for each feature within a given feature group are equal. This approach is overly simplistic and fails to reasonably capture the differences in importance among the various features.
2.	In section 4, the author highlights an issue where variations in appearance frequency can lead to an unfair assessment of feature importance when the sample size N is small. However, there is no mathematical analysis of Eq. (9) to demonstrate how the designed importance score addresses this issue.
3.	In section 5.1, why not limit k < |S| instead of considering the special case that |S| < k.
4.	The importance score is calculated based on the frequency with which a feature is selected and the predicted label, meaning that two features that are occasionally selected together end up with the same importance score. In contrast, Shapley value calculations based on label probability would differentiate between these features. Consequently, the proposed ENSEMBLESHAP, which relies on this importance score, assigns identical values to these features, potentially overlooking the differences in their individual influences.
5.	The authors claim that the proposed method is computationally efficient. However, there is a lack of analysis regarding its complexity and the associated time costs.

**Questions:**

Please help to check the weaknesses part.

---

> ### Author Response · Authors · 2024-11-21
>
> We thank the reviewer for the valuable feedback. We address the questions below:
>
> **Weakness 1. In section 4, the importance scores for each feature within a given feature group are equal. This approach is overly simplistic and fails to reasonably capture the differences in importance among the various features.**
>
> We note that these feature groups are randomly sampled and can be vary. In Section 5.3, we demonstrate that our method satisfies the essential properties of effective feature attribution and is order-consistent with the Shapley value. For more details, please refer to Weakness 4.
>
> **Weakness 2. In section 4, the author highlights an issue where variations in appearance frequency can lead to an unfair assessment of feature importance when the sample size N is small. However, there is no mathematical analysis of Eq. (9) to demonstrate how the designed importance score addresses this issue.**
>
> Thank you for pointing that out. We conduct an empirical comparison between two approaches: (1) directly applying Monte Carlo sampling, and (2) Monte Carlo sampling with normalization based on appearance frequency (Eqn. 9). Our experiments focus on adversarial attacks, with $N$ set to 200. The results demonstrate that normalizing the importance scores by appearance frequency improves the performance.
>
> | Dataset | SST-2 | IMDB | AG-news |
> |-----------------|-----------------|-----------------|-----------------|
> | Without Normalization | 0.82    |  0.95  |  0.92   |
> | With normalization  | 0.87    | 0.99   | 0.96    |
>
>
> **Weakness 3. In section 5.1, why not limit k < |S| instead of considering the special case that |S| < k.**
> In Section 5.1, we follow the standard definition of the Shapley value, which is calculated as the average of marginal contributions across all feature subsets, ranging in size from $0$ to $d-1$. This implies that for any given feature subset $S$, regardless of its size, the ensemble model should always be able to make predictions.
>
> However, by the original definition, the ensemble model cannot subsample $k$ features from the provided $|S|$ features if $|S| < k$. To address this limitation and ensure theoretical rigor, we mention this special case in our analysis.
>
> **Weakness 4. The importance score is calculated based on the frequency with which a feature is selected and the predicted label, meaning that two features that are occasionally selected together end up with the same importance score. In contrast, Shapley value calculations based on label probability would differentiate between these features. Consequently, the proposed ENSEMBLESHAP, which relies on this importance score, assigns identical values to these features, potentially overlooking the differences in their individual influences.**
>
> Thank you for the question. In fact, two features that are occasionally selected together will not have the same importance score. They will only have identical scores if they are always selected together in every subsampled feature group. However, the probability of this occurring decreases exponentially as the number of subsamples increases.
>
> As we prove in Section 5.3, our method is order-consistent with the Shapley value when a large number of feature groups are subsampled. In other words, if the Shapley value can differentiate between these features, our method can do so as well.
>
> **Weakness 5. The authors claim that the proposed method is computationally efficient. However, there is a lack of analysis regarding its complexity and the associated time costs.**
>
> Given that the random subspace method is already deployed, our method introduces negligible additional computational time (less than 3 seconds) for the explanation.

---

> > ### Comment · Reviewer_52Co · 2024-11-26
> >
> > I acknowledge that I have read the response.

---

> > > ### Author Response · Authors · 2024-12-02
> > >
> > > We hope our explanation adequately addresses your concerns and encourages you to reconsider the value of our paper. We are happy to address any further questions or concerns you may have. Thank you once again for your time and dedication in reviewing our work.

---

### Official Review · Reviewer_2GZo · 2024-11-04

**Soundness:** 3
**Presentation:** 2
**Contribution:** 2
**Rating:** 6
**Confidence:** 3

**Summary:**

The paper presents EnsembleSHAP, a novel feature attribution method tailored for the random subspace method. EnsembleSHAP addresses limitations in traditional feature attribution approaches, such as Shapley values and LIME, which are computationally intensive and lack security assurances against explanation-preserving attacks. EnsembleSHAP leverages computational byproducts of the random subspace method to provide efficient, accurate, and secure explanations for model predictions. This method is specifically designed to improve resilience against adversarial and backdoor attacks, as well as jailbreaking attacks on large language models. Experimental results show that EnsembleSHAP outperforms baseline attribution methods in identifying harmful features under various security threats, including certified defense and jailbreaking scenarios. The theoretical analysis demonstrates that EnsembleSHAP maintains key properties of effective feature attribution, such as local accuracy and robustness against attacks.

**Strengths:**

1. The paper is structured logically, moving from the problem context and related work to problem formulation, method design, theoretical analysis, and empirical validation.

2. The authors provide a theoretical basis for EnsembleSHAP.

3. EnsembleSHAP leverages the computational byproducts of random subspace methods, resulting in lower computational overhead compared to traditional methods.

4. The paper considers multiple threats - adversarial attack, backdoor attack, and jailbreaking.

**Weaknesses:**

1. EnsembleSHAP is designed specifically for random subspace methods, which could limit its generalizability to other ensemble methods or broader feature attribution applications that do not involve subsampling.

2. The efficiency claim is not well studied in the experimental section.

3. The certified detection theorem and detection strategy are not clearly explained, making it difficult for readers to fully understand the approach and its guarantees.

4. The method’s assumptions about limited modifications to input features may not hold for many real-world backdoor attacks, where an attacker might poison the entire input space or apply more complex poisoning strategies. This assumption restricts the generalizability of the certified detection method for a wider range of attacks.

5. The paper evaluates EnsembleSHAP using TextFooler for adversarial attacks and BadNets for backdoor attacks.These attacks are somewhat dated, and there are newer, more sophisticated adversarial and backdoor attacks in current literature. Testing against more recent attacks could better demonstrate the robustness of EnsembleSHAP. In fact, one can even design an adaptive attack.

**Questions:**

My major questions are included in the above weaknesses comments.

---

> ### Author Response · Authors · 2024-11-21
>
> We thank the reviewer for the constructive comments.
>
> **Weakness 1.EnsembleSHAP is designed specifically for random subspace methods, which could limit its generalizability to other ensemble methods or broader feature attribution applications that do not involve subsampling.**
>
> Random subspace methods have a wide range of applications. In this paper, we focus on their use for defending against attacks in the input space. Additionally, random subspace methods have been applied to differential privacy (Liu et al., 2020), facilitate machine unlearning (Bourtoule et al., 2021), and build robust models to withstand poisoning attacks (Jia et al., 2021). Exploring the potential of EnsembleSHAP in these and other applications of random subspace methods represents an intriguing direction for future research.
>
> **Weakness 2. The efficiency claim is not well studied in the experimental section.**
>
> Our method serves as a post-hoc explanation technique for the random subspace method, leveraging computational byproducts while introducing negligible costs (less than 3 seconds). Improving the computational efficiency of the random subspace method itself remains an open challenge (Jia et al., 2021;Zhang et al.,2023; Zeng et al.,2023).
>
> **Weakness 3. The certified detection theorem and detection strategy are not clearly explained, making it difficult for readers to fully understand the approach and its guarantees.**
>
> Thank you for pointing that out. The explanation of the certified detection theorem is brief due to space constraints. The theorem states that if an attacker modifies $T$ features of the original testing input $x$ to alter the predicted label of the ensemble classifier, our method can guarantee that $D(x, T)$ of these modified features will be detected as the most important features.
>
> To provide some intuition, consider an extreme case where the subsampling size $k$ is 1, meaning each feature gets a single vote. Suppose there are 5 features in total, and all features initially vote for the correct label before the attack. To flip the predicted label, the attacker would need to change the predictions of at least 3 features to the target label. In this scenario, the contributions of these 3 modified features to the target label would surpass those of the unmodified features (which contribute nothing to the target label). Therefore, if we report top-3 most important features for the predicted label (after the attack), these 3 modified features are provably reported.
>
> The guarantee $D(x, T)$ depends on several factors in the general case:
>
> 1. Confidence of the ensemble model's prediction before the attack:  If the ensemble model is highly confident, the modified features must influence a greater number of subsampled groups to alter the prediction, making them more detectable.
>
> 2. Subsampling ratio:  A smaller subsampling ratio enhances certified detection performance.
>
> 3. $T$:  A larger $T$ makes certified detection more challenging.
>
> These factors collectively impact the effectiveness of the certified detection.
>
> **Weakness 4. The method’s assumptions about limited modifications to input features may not hold, where an attacker might poison the entire input space or apply more complex poisoning strategies.**
>
> Our certified detection guarantee holds for any complex poisoning strategies, provided the number of poisoned features is bounded. However, we acknowledge that certified detection becomes increasingly difficult as the number of perturbed features grows. This is because adversarial features can collectively influence the target label, reducing the individual contribution of each adversarial feature to a level that becomes indistinguishable from benign features (which may also inadvertently contribute to the target label). It is worth noting that this challenge also exists in certified defenses. Indeed, existing certified defenses (Jia et al., 2021; Zhang et al.,2023; Zeng et al.,2023) cannot certify for >10% of adversarial features. In practice, our method is still effective when 20% of the input features are poisoned (in adversarial attacks).
>
> **Weakness 5. Some attacks evaluated are somewhat dated, and there are newer, more sophisticated adversarial and backdoor attacks. In fact, one can even design an adaptive attack.**
>
> We employ TextFooler for adversarial attacks and BadNets for backdoor attacks because they are representative and can be readily applied to ensemble models. Many advanced white-box adversarial attacks require substantial modifications and high computational costs to adapt to ensemble models. For black-box attacks, the design of state-of-the-art methods fundamentally aligns with TextFooler, as they all rely on a trial-and-error process for black-box optimization. Additionally, our certified detection analysis indicates that the effectiveness of our method depends largely on the total number of modified features, rather than the specific attack strategy employed.

---

> > ### Comment · Reviewer_2GZo · 2024-11-27
> > **Follow-up questions**
> >
> > Thank you for your response to my previous questions. I have two follow-up questions:
> >
> > 1. The first question is regarding your design choice in EnsembleSHAP. Specifically, the method uses the indicator function to attribute feature importance, rather than utilizing the class probabilities computed by the model. Using the indicator function will potentially overestimate or underestimate certain features. Could you clarify why the indicator function was chosen over class probabilities?
> >
> > 2. The second question is regarding the practicality of the detection mechanism. While EnsembleSHAP offers certified detection of adversarially modified features, it seems that this detection is only effective if the defender already knows that an attack has occurred. In practical scenarios, however, defenders may not have prior knowledge of an attack and might rely on mechanisms to detect anomalous or adversarial inputs before applying feature attribution methods. Could you clarify how EnsembleSHAP could be used in scenarios where the defender does not know whether the data has been attacked? Does the method rely on external tools or assumptions, such as baseline comparisons or anomaly detection, to identify suspect inputs for analysis? If so, how might these external mechanisms interact with EnsembleSHAP to form a complete detection pipeline?

---

> > > ### Author Response · Authors · 2024-11-28
> > >
> > > Thank you for your thoughtful reply. We have addressed the questions as follows:
> > >
> > > **Why use indicator function instead of class probabilities?**
> > >
> > > We use an indicator function in our method to ensure its applicability to a wider range of classifiers, including black-box classifiers that do not output class probabilities. For example, consider the application of our method to RA-LLM, a defense mechanism against jailbreaking attacks. The base classifier in RA-LLM is an alignment-check function that predicts the label "harmful" if the LLM's output (based on subsampled input text) includes phrases such as "I am sorry, but I cannot answer this ...", and predicts "non-harmful" otherwise. RA-LLM then aggregates these predictions across all subsampled texts to make the final decision (e.g., whether to reject or not). In such cases, using an indicator function allows our method to be more broadly applicable.
> > >
> > > **Practicality of the detection mechanism. Could you clarify how EnsembleSHAP could be used in scenarios where the defender does not know whether the data has been attacked? Does the method rely on external tools or assumptions, such as baseline comparisons or anomaly detection, to identify suspect inputs for analysis? If so, how might these external mechanisms interact with EnsembleSHAP to form a complete detection pipeline?**
> > >
> > > The detection mechanism of EnsembleSHAP can be used as an post-attack forensic analysis tool within a defense pipeline. Specifically, a defender can first employ prevention or detection-based methods, such as anomaly detection, to identify whether an attack has occurred. Once an attack is detected, EnsembleSHAP can then be utilized to trace the root cause of the attack. Notably, even without integration with external tools, EnsembleSHAP can still be directly applied as a reliable explainer for the outputs of the ensemble model.

---

> > > > ### Comment · Reviewer_2GZo · 2024-12-02
> > > > **Response**
> > > >
> > > > Thank you for your response. The authors have addressed most of my concerns (although I am still not fully convinced about the robustness aspect). I have increased my score accordingly.

---

> > > > > ### Author Response · Authors · 2024-12-02
> > > > >
> > > > > Thank you for provide additional feedback and share your concerns with us. We appreciate your thoughtful review and recognition of our efforts.

---

### Official Review · Reviewer_bQ2B · 2024-11-04

**Soundness:** 3
**Presentation:** 2
**Contribution:** 3
**Rating:** 3
**Confidence:** 3

**Summary:**

This paper proposes EnsembleSHAP, a feature attribution method based on the well-known random subspace method, which is claimed to be computationally efficient and preserves fundamental properties of Shapley values. The method provides certifiable robustness against explanation-preserving attacks to language models, as theoretically shown.

**Strengths:**

- Relevant topic;
- Theoretical analysis;
- Rich model and attack types considered in the experiments.

**Weaknesses:**

- Unclear presentation;
- No empirical evaluation of the computational complexity;
- Lack of comparison with other efficient feature attribution methods;
- The proposed algorithm is not formally stated but only described verbally.

**Comments.**

**Unclear presentation.** Presentation needs substantial improvement. One unclear point to me is that the random subspace method proposed by T. K. Ho does not work in the way it is used in this paper, as far as my understanding of this work is concerned. The random subspace method creates distinct training sets by bagging and subsampling the feature set in each round, and it's the basic method used to train random forests. I don't see how it is directly applied in this work (at least, it's unclear how it's applied at training vs test time). It was originally proposed to boost the performance of classifier ensembles, and it had nothing to do with security issues. This should also be clarified in the paper, I guess that it's only the recent developments that used that method to get certified robustness via randomization (a la randomized smoothing).

**Lack of empirical computational complexity analysis.** The authors did not provide any evaluation of the computational complexity required for computing the importance scores with the proposed method, nor they provided information on what algorithm they used for estimating the standard Shapley values. I don't buy that this method is computationally efficient, if it requires sampling as many as 10,000 different inputs before providing a prediction.

**No comparison with other efficient Shapley values estimation techniques.** Other efficient methods have been previously proposed for efficient Shapley values estimation; despite this, the authors did not provide a comparison with other methods, e.g., FastSHAP [1].

**Formal algorithm is missing.** In Sect. 4, there is no actual definition of the algorithm. Instead, a description of the used methods is given in words, such as “Monte Carlo” sampling or the approximation of the defined importance score. The approach to solving the presented optimization problem has not been reported.

**No further discussion of the certified detection rate results.** In Sect. 6.3 the plot of the certified detection rate against the top-e important features is reported. However, there is no discussion on the obtained results; there is no discussion on the total number of considered features, why the detection reaches a plateau after few “e”. This require further elaboration.
Moreover, the experiments on jailbreaking, the motivation behind the choice of the hyperparameters, and other relevant experiments are confined in the appendix. The authors should reconsider that to make the paper more self-contained.

[1] Jethani, N., Sudarshan, M., Covert, I. C., Lee, S.-I., & Ranganath, R. (2022). FastSHAP: Real-Time Shapley Value Estimation. International Conference on Learning Representations. Retrieved from https://openreview.net/forum?id=Zq2G_VTV53T

**Questions:**

1. Why did the authors not provide a comparison with other efficient methods for Shapley values estimation, like FastSHAP [1]? Is it because they are still inefficient when applied to random subspace methods?

2. What approach is used for solving the optimization problem stated in Sect. 5.4?

3. What is the rationale behind selecting the ICL [2] method as a baseline? How did the authors adapt it to work as a feature attribution method?


[2] Nicholas Kroeger, Dan Ley, Satyapriya Krishna, Chirag Agarwal, and Himabindu Lakkaraju. Are large language models post hoc explainers? arXiv preprint arXiv:2310.05797, 2023

---

> ### Author Response · Authors · 2024-11-21
>
> Thanks for the feedback!
>
> **Weakness 1. Unclear presentation.**
> The general approach of bagging or subsampling the feature set was, to the best of our knowledge, originally introduced to enhance the performance of decision trees. Therefore we follow the naming. We will provide additional clarification of this terminology in our paper.
>
> **Weakness 2. Lack of empirical computational complexity analysis.**
> We note that our method serves as a post-hoc explanation for the random subspace method. This means the defender first makes predictions using the random subspace method (which involves a large number of base model queries to improve robustness), and then our method provides an explanation for those predictions. Our key claim is that our approach incurs negligible additional computational cost (less than 3 seconds to compute Eqn.9), as it reuses the computational byproducts already generated during the deployment of the random subspace method. In contrast, other feature attribution methods require additional computation time because they are not specifically tailored to the random subspace method.
>
> **Weakness 3. No comparison with other efficient Shapley values estimation techniques.**
> Given that the RSM is already deployed, our method introduces negligible additional computational overhead (less than 3 seconds). FastSHAP can be seen as an extension of LIME, which needs to build up a training dataset for the explainer model. When applied directly to the random subspace method, it needs significant additional computational time.
>
> **Weakness 4. The formal algorithm is missing.**
> Sorry for the confusion. Eqn.9 in Section 4 provides the formal calculation of our method with “Monte Carlo” sampling. We use binary search to solve the presented optimization problem in Section 5.4. We will provide clarification on this in our paper.
>
> **Weakness 5. No further discussion of the certified detection rate results. The paper should be more self-contained.**
> Thank you for pointing this out. The explanation for the experimental results on the certified detection rate was short due to space constraints. From our experiments, we observed that the certified detection rate is insensitive to the total number of features (e.g., IMDB has much longer input sequences than AG-News, but both exhibit similar certified detection rates). However, the certified detection rate is strongly influenced by the ratio of subsampled features to the total number of features, as shown in Figure 17 in the Appendix. Specifically, a smaller subsampling ratio (or larger dropping ratio) improves the certified detection rate. This occurs because each adversarial feature influences fewer subsampled groups, making it harder for these features to change the predicted label without being detected.
>
> Regarding the plateau observed after a few "e," we notice a turning point when "e" reaches the number of adversarial features "T." Beyond this point, the number of reported features "e" is no longer the bottleneck. Instead, the certified detection rate depends more on "T." When "T" is large, in the worse case, the attacker can distribute the influence of perturbed features more evenly toward the target label, making each adversarial feature contribute less to the target label. Since there are benign features that may inadvertently contribute to the target label, these perturbed features become hard to detect.
>
> Some experimental details are in the appendix because of space reasons. We will further refine our paper.
>
> **Question 1. Why did the authors not provide a comparison with other efficient methods for Shapley values estimation, like FastSHAP [1]?**
> Please refer to Weakness 3 for details. In summary, our method repurposes the computational byproducts generated during the ensemble model's predictions to provide explanations, with no additional computational cost. Additionally, we theoretically demonstrate that our explanation is order-consistent with the Shapley value for the ensemble model, which is hard to compute.
>
> **Question 2. What approach is used for solving the optimization problem stated in Sect. 5.4?**
>
> We use binary search to solve the optimization problem in Section 5.4. We will provide clarification on this in our paper.
>
> **Question 3. What rationale is behind selecting the ICL as a baseline? How is it adapted for feature attribution?**
>
> We think it would be interesting to leverage the in-context learning capabilities of large language models (LLMs) to build the explainer, as LLMs are becoming increasingly powerful and widely adopted. For this purpose, we use the GPT-3.5-turbo model. The prompt includes an in-context learning dataset, where the inputs consist of indexes of subsampled features, and the labels correspond to the predictions of the base model based on these subsampled features. The prompt includes an instruction guiding the LLM to output a ranked list of a certain fraction (e.g.,20%) of the most important features.

---

> > ### Comment · Reviewer_bQ2B · 2024-12-02
> > **Acknowledge**
> >
> > Thank you for your response, which clarified a few issues. However, the lack of comparison with competing approaches and presentation clarity remain unaddressed. I will thus keep my score and invite the authors to address these points to improve their paper.

---

### Meta-Review · Area_Chair_CSpi · 2024-12-20

**Metareview:**

This work proposed a post explanation method for random subspace method, and provided certified defenses against several attacks (adversarial examples, backdoor attacks, jailbreaking attacks), with the benefit of reduced computational cost.

It received 4 detailed reviews. The strengths mentioned by reviews mainly include the theoretical analysis, the reduced cost by utilizing the computational byproducts of random subspace methods, several threats.

Meanwhile, there are also several important concerns, mainly including the lack of comparison with other efficient methods, lack of empirical computational complexity analysis, the certified detection theorem and detection strategy are not clearly explained, outdated attacks, the assumption about limited modifications to input features may not hold, and experiments (adversarial and backdoor attacks against LLM, adaptive attacks).

The authors provided rebuttals for these concerns, and some reviewers gave further feedback. Generally speaking, the authors didn't made enough efforts to address these concerns, for example, several suggested experiments are not followed. Several important concerns are not well addressed. Thus, my recommendation is reject.

**Additional Comments On Reviewer Discussion:**

The rebuttal and discussions, as well as their influences in the decision, have been summarized in the above metareview.

---

### Decision · Program_Chairs · 2025-01-22

Reject